# Integrative Medicine Focusing on Ultrasound-Guided High-Dose Shinbaro 2 Pharmacopuncture for Acute Herniated Intervertebral Discs: A Case Report

**DOI:** 10.3390/healthcare12161620

**Published:** 2024-08-14

**Authors:** Nam-Woo Lee, Jinho Lee, Jung-Min Yun, Siwon Kim, Won-Seok Chung

**Affiliations:** 1Department of Clinical Korean Medicine, Kyung Hee University, 23 Kyunghee-daero, Dongdaemun-gu, Seoul 02447, Republic of Korea; kaktus0831@jaseng.co.kr; 2Jaseng Hospital of Korean Medicine, 536 Gangnam-daero, Gangnam-gu, Seoul 06110, Republic of Korea; jasengjsr@gmail.com (J.L.); dbsals0622@jaseng.co.kr (J.-M.Y.); toy0993@jaseng.co.kr (S.K.)

**Keywords:** Shinbaro 2 pharmacopuncture, herniated intervertebral disc, pain, Korean medicine, ultrasound-guided pharmacopuncture

## Abstract

This study aimed to investigate the effects of ultrasound-guided high-dose Shinbaro 2 pharmacopuncture on the pain, dysfunction, and quality of life in patients with low back pain and radiating pain due to an acute herniated intervertebral disc (HIVD). A 39-year-old male patient with low back pain and radiating pain caused by an acute HIVD was treated with Korean and Western integrative medicine, with a focus on ultrasound-guided high-dose Shinbaro 2 pharmacopuncture at Kambin’s triangle. The treatment lasted 16 weeks, including a 12-day hospitalization. The low back pain and radiating pain were evaluated using the numeric rating scale (NRS). The lumbar function and quality of life were assessed using the Oswestry disability index (ODI) and the EuroQol five-dimension index (EQ5D). Satisfaction was gauged using the patient global impression of change (PGIC). After treatment, the NRS score decreased from 10 to 1, whereas the ODI and EQ5D scores improved from 84.44 to 28.89 and from 0.303 to 0.871, respectively. The PGIC was rated as 1, indicating considerable improvement. Notably, the changes observed during hospitalization were significant. This report suggests that ultrasound-guided high-dose Shinbaro 2 pharmacopuncture at Kambin’s triangle significantly improves the pain, dysfunction, and quality of life in patients with an acute HIVD, demonstrating its potential usefulness among Korean medicine practitioners.

## 1. Introduction

The human lumbar spine is formed by the alternating placement of vertebral bodies and intervertebral discs (IVDs). Under normal conditions, the annulus fibrosus (the outer shell of the lumbar disc) consists of layers of collagen protein fibers, and the nucleus pulposus (a softer type of collagen protein) fills the central portion of the lumbar disc. The annulus fibrosus may be torn when an external force or pressure is applied to the IVDs, resulting in the displacement of the nucleus pulposus inside. In severe cases, some of the nucleus pulposus from the disc core may leak out; this condition is referred to as a “herniated intervertebral disc” (HIVD). The leaked nucleus pulposus compresses the spinal dura or nerve root, resulting in localized pain in the area adjacent to the HIVD, radiating pain, hypoesthesia (numbness), and muscle weakness in the area innervated by the compressed nerve [1,2]. Previous studies on disc-related pathology have shown that lumbar disc herniation (LDH) is a major cause of low back pain (LBP) and sciatica [3] and contributes to the increasing socioeconomic burden of non-fatal diseases globally [4,5].

For the diagnosis of LDH, magnetic resonance imaging (MRI) is recommended as the gold standard for confirming the presence of a lesion in the herniated lumbar disc, which is the source of recurrent pain. Among the different types of LDH, patients with the extrusion type of disc herniation complain of radiculopathy spreading to the leg, as well as LBP. Surgical interventions and conservative therapies, such as a selective nerve root block (epidural injection), pain relievers, and physical therapy/exercise, are used in clinical practice [6]. Conservative treatment refers to non-surgical methods used to manage and treat medical conditions, including an acute HIVD. These treatments are aimed at reducing symptoms, improving function, and promoting recovery without the need for invasive procedures [7]. A nerve block (lumbar epidural injection) is a representative conservative treatment associated with complications such as accidental injury to other nerves or vessels [8]. Thus, a nerve block is performed under conventional computed tomography or C-arm fluoroscopic guidance to avoid complications and to allow for an accurate and safe approach, thereby improving the procedural success. However, these types of modalities have drawbacks, including exposure to radiation for both the patients and the practitioners and the relatively high cost of service, and they can only be used in a limited number of institutions equipped with specific settings [9]. To overcome these limitations, the use of ultrasound for pain treatment has become increasingly popular owing to its advantages, such as its ease of access, low barrier of equipment involved, and absence of ionizing radiation exposure [10]. Ultrasound-guided spinal interventions for LBP are mainly administered in the facet joints, medial branches, nerve roots, and epidural space [11]. Nonetheless, this ultrasound-guided technique directly applied to the IVDs and nerve roots through Kambin’s triangle has not yet been commonly used as a general intervention approach.

Given the limitations of conservative treatments based on conventional (Western) medicine, conservative treatments based on complementary medicine, such as acupuncture, pharmacopuncture, and herbal medicine, have attracted growing interest, and reports on the effectiveness of these treatments have been actively published [12]. Among the various complementary medicine modalities, pharmacopuncture is a Korean medicine treatment involving the injection of natural products or herbal medicines at various acupoints [13]. In pharmacopuncture administered for the treatment of spinal diseases, a small amount of pharmacopuncture solution is injected at acupoints around the affected area of the spine. The effectiveness of pharmacopuncture for LBP has been demonstrated in randomized controlled trials of pharmacopuncture as an intervention [14]. In the clinical practice of Korean medicine, pharmacopuncture for LDH is actively used and has received a grade B recommendation (moderate level of evidence) in the Korean Medicine Clinical Practice Guidelines for Lumbar Herniated Intervertebral Disc [15]. Considering the clinical benefits, such as the increased efficacy of pharmacopuncture and a faster recovery if the procedure is administered close to the lesion area at a high dose to ensure accuracy and safety, research has been conducted on the method of developing the current practice of pharmacopuncture administration. In this context, ultrasound-guided pharmacopuncture has recently emerged. Furthermore, the administration of a high-dose pharmacopuncture solution to increase the efficacy of pharmacopuncture for the treatment of spinal diseases has been suggested. In the current clinical practice of traditional Korean medicine, megadose pharmacopuncture is administered without guidance [16]. Previous research has explored ultrasound-guided acupotomy and herbal medicine for LDH [17], but no previous studies have specifically examined the treatment effects of ultrasound-guided high-dose Shinbaro 2 pharmacopuncture in patients with LDH, to the best of our knowledge. This study aimed to fill that gap by investigating the efficacy and safety of this novel approach.

Herein, we report a case of a patient with an acute HIVD complaining of LBP for whom ultrasound guidance was used for the Kambin’s triangle approach to administer high-dose pharmacopuncture. The patient showed a remarkably rapid improvement, suggesting that the proposed method could serve as a new effective approach for pharmacopuncture. This case report describing a single patient was exempted from our hospital’s Institutional Review Board (IRB) approval (JASENG 2023-10-003). It was prepared in compliance with the case report guidelines (CARE guidelines) [18] (Appendix A).

## 2. Case Presentation

### 2.1. Clinical Feature

The patient was a 39-year-old Asian man who worked in the agricultural sector. On 14 February 2023, during his farming activity, he sprained his back while stepping on the ground as he was getting off a tractor, resulting in the onset of LBP, as well as numbness and a tingling sensation that spread to the anterior and lateral sides of his left calf (low back pain in the lower extremity: +, numbness: +, tingling sensation: +). On 15 February, he visited a local orthopedic surgery clinic and was prescribed oral analgesics; however, his symptoms did not improve. On 16 February, he visited the Rehabilitation Department of Traditional Korean Medicine at our hospital. There is no relevant medical or family history associated with this clinical presentation.

### 2.2. Physical Examination

The patient’s pain was aggravated in the sitting and bending positions. Additionally, the pain intensity increased when walking, making it impossible for the patient to walk for more than 10 min.
-Range of motion: flexion, 20; extension, 20; lateral bending, 30/30; rotation, 20/20.-Special test (R/L): straight leg raise test, 60/40; Patrick’s test, −/−; femoral nerve stretch test, −/−.-Deep tendon reflex: patellar, ++/++; Achilles, ++/++.-Manual muscle testing: dorsiflexion, (G5, 100%)/(G5, 100%); big toe extension, (G5, 100%)/(G5, 100%); plantar flexion, (G5, 100%)/(G5, 100%).-Sensory: hypoesthesia (numbness) on the posterolateral side of the left thigh and on the anterolateral side of the left calf (L5 and S1 dermatomes).

### 2.3. Radiological Examination

On lumbar spine T2-weighted turbo-spin-echo MRI conducted on 18 February 2023, disc extrusion from the central to left foraminal zone at the L4–L5 spinal segment was observed with a high signal intensity in the annular fissure. Additionally, the left L4 and proximal left L5 nerve roots were closely abutted against the extruded disc (Figure 1).

### 2.4. Diagnosis

The patient, an agricultural worker, had a clear cause of exacerbated pain two days prior to the initial consultation. Based on the findings from the physical examination, which included a restricted straight leg raise of 60/40, a limited lumbar spine range of motion, symptoms along the L5-S1 dermatome, and L4/5 disc extrusion observed on the lumbar spine MRI, the patient was diagnosed with acute herniated intervertebral discs. This diagnosis corresponds to ICD-10 code M51.1 for “lumbar and other intervertebral disc disorders with radiculopathy”.

### 2.5. Treatment

#### 2.5.1. Treatment Period

The patient visited the hospital for an initial consultation on 16 February. He received inpatient treatment for 12 days from 17–28 February, followed by outpatient treatment until 7 June. After discharge, outpatient treatment was performed once a week for a total of 16 weeks. Table 1 summarizes the treatment methods employed according to the timeline.

#### 2.5.2. Ultrasound-Guided Pharmacopuncture

Ultrasound-guided pharmacopuncture was performed by placing one or two pillows under the patient’s abdomen and positioning the patient to reduce lordosis while securing some space behind the patient. To prevent infection, the practitioner wore latex gloves, disinfected the skin with a povidone–iodine solution, and adopted the complete aseptic technique throughout the procedure. A 26-gauge needle, which was relatively fine and caused minimal tissue damage, was used for the procedure. Additionally, the slight tingling sensation typically felt during pharmacopuncture is generally within the tolerable range for patients. Therefore, anesthesia was not administered during the treatment.

Scanning was performed at 4.6 MHz using the ultrasound probe of a convex array transducer (Alpinion Medical Systems). Prior to the ultrasound examination, the condition of the IVD was observed, and the target area and level for the procedure were accurately located and measured by analyzing the MR images. The concept of “Kambin’s triangle”, formed by the exiting nerve root from the segment, the superior border of the caudal vertebra, and the dura/traversing nerve root, was employed to ensure a safe and effective approach to the IVD and the nerve root of the applicable spinal segment as closely as possible (Figure 2).

For long-axis scanning of the lumbar spine, the probe was placed on a palpable spinous process at the level of the posterosuperior iliac spine. Subsequently, the probe was moved laterally to identify the appearance of the camel-hump sign, which was determined to be the facet joint (Figure 3A). Next, the probe was moved cranially from the sacrum lying longitudinally, and the vertebral segments L5/S1 and L4/L5 were located and marked according to the order of the joints observed in this direction. From the long-axis view, the target segment for intervention (i.e., the L4–L5 facet joint) was determined, and the probe was rotated by 90° to perform short-axis scanning (Figure 3B). In the short-axis view, with the spinous process placed at the center, the probe was moved so that the superior articular process of the facet joint was displayed at the center of the screen to obtain a paramedian view. The probe was moved caudally from this position to scan the L5 transverse process (Figure 3B,C). Afterwards, the probe was moved cephalad, and the transverse process disappeared; the location for the intervention was determined in this plane (Figure 3D).

After locating the target area for intervention, 6 mL of Shinbaro 2 pharmacopuncture solution was injected using a 4-inch 26-gauge needle through an in-plane approach under ultrasound guidance with the probe (Figure 4A, Table 1). The direction of needle insertion pointed to the anterior side of the superior articular process, and the pharmacopuncture solution was injected slowly while visually checking the direction of the needle on the screen to ensure that the patient did not feel severe pain during the process (Figure 4B). When the needle tip reached the level of the superior articular process, it was first checked to ensure that it did not touch the bone, and the needle was then advanced for an additional 0.3–0.5 cm in the anteromedial direction of the superior articular process, with the needle tip disappearing from the screen. Additionally, the end-feel of the intradiscal space where the needle was located was soft and elastic, ensuring that no notable resistance occurred when in contact with the bony structure or during injection. The needle was advanced with power Doppler imaging to prevent a radicular artery puncture. The advancement of the needle was stopped immediately when the patient felt an electric shock sensation, which indicated nerve root irritation caused by the needle. The pharmacopuncture solution (usually 6 mL) for the treatment of compressed nerve roots was injected into both the peridiscal and perineural spaces until a great resistance was felt, and the patient was monitored to ensure that he felt pain of a constant intensity without it becoming more severe during the injection. During the procedure, the patient experienced localized tightness and heaviness at the initial drug injection site, and as the amount of drug increased, a tightening and heavy sensation gradually developed along the L5 dermatome. Following the completion of the procedure, the patient was instructed to lie in the prone position for 20 min and to wear a brace immediately after the procedure. The symptoms experienced during the procedure completely dissipated 2–3 h after the procedure.

Pharmacopuncture administration commenced after admission and was performed every other day, except for 27 and 28 February, during the 12-day inpatient treatment period, for a total of six sessions. The first session, conducted on 17 February, utilized the Kambin’s triangle approach, based on clinical symptoms, to target the L4/L5 region prior to the MRI on 18 February. Subsequent sessions, following MRI confirmation of the L4/L5 disc lesion, continued to use the Kambin’s triangle approach for the precise treatment of the affected area. After discharge, pharmacopuncture was administered once weekly at each visit during the outpatient treatment period, for a total of 16 sessions (Table 1).

#### 2.5.3. General Pharmacopuncture

Ashi points mainly around the muscles showing tension, such as the quadratus lumborum and erector spinae muscles, were selected. For each acupoint, 1 mL of Shinbaro 2 pharmacopuncture solution was injected using a 0.33 mm (29 G) × 13 mm insulin syringe (1 mL/cc; SUNGSHIM Medical, Bucheon, Republic of Korea) (Table 2). Regarding outpatient treatment, general pharmacopuncture was performed once on 16 February, for a total of 17 sessions from discharge to 7 June. During the inpatient treatment period, it was administered twice a day in the morning and afternoon (Table 2).

#### 2.5.4. Acupuncture

Acupuncture was performed using disposable, sterile, stainless-steel filiform needles (0.25 × 40 mm; Dong Bang Medical, Boryeong, Republic of Korea), which were retained for 15 min. With respect to the local acupoints, bilateral Shenshu (腎兪; BL23), Qihaisu (氣海兪; BL24), Dachangshu (大腸兪; BL25), Guanyuanshu (關元兪; BL26), Weizhong (委中; BL40), Zhishi (志室; BL52), Huantiao (還跳; GB30), Yanglingquan (陽陵泉; GB34), Xuanzhong (懸鍾; GB39), and Hwatahyeopcheok (夾脊穴) points were mainly selected. The typical depth of needle insertion was 20–30 mm. Infrared therapy (INFRALUX-300; Daekyung Electro Medical, Pocheon, Republic of Korea) was also administered during needle retention. The frequency and duration of the acupuncture sessions were the same as those of the general pharmacopuncture sessions (Table 2).

With a low-frequency electrical stimulator (STN-111; StraTek, Anyang, Republic of Korea), wire lines were connected to the above acupoints to perform electroacupuncture at a frequency of 2 Hz and at an intensity for which the patient could slightly sense the stimulation. Electroacupuncture was administered concurrently with the acupuncture (Table 2).

#### 2.5.5. Chuna Manual Therapy

For the simple Chuna technique, the supine cervical JS distraction correction technique, the side-lying lumbar “pitch and roll” distraction method, and the occipital correction technique were employed. For complex Chuna techniques, the supine cervical correction technique, the prone both pisiform lower thoracic flexion displacement correction technique, the side-lying lumbar flexion displacement correction technique, the prone posteriorly rotated ilium correction technique/sacrum lateral flexion correction technique, and the prone sacrum lateral flexion rotated displacement correction technique were used. During the inpatient treatment period, Chuna manual therapy was performed once daily, except for 25 and 26 February, for a total of 10 sessions. As for outpatient treatment, Chuna manual therapy was administered 16 times from discharge to 7 June, excluding the day of the first visit (Table 2).

#### 2.5.6. Herbal Medicine

During the inpatient treatment period, Cheongpajeon and Shingyeongbaro-hwan (Shinbaro herbal medicines) and Bunsosan (an herbal medicine used for indigestion) were administered three times daily. At 1 month after discharge, the two herbal medicines, excluding the gastrointestinal agent, were administered twice daily (Table 2).

#### 2.5.7. Physical Therapy

During the inpatient treatment period, ultrasonography, interferential current therapy, and laser therapy were performed eight times. For outpatient treatment, the same types of physical therapy were performed 16 times from discharge to 7 June, excluding the day of the first visit. In the case of traction therapy, pelvic traction therapy was performed twice (on 3 and 10 April) (Table 2).

#### 2.5.8. Pharmacological Treatment (Western Medicine)

On the day of the first visit, oral analgesics were administered once as outpatient treatment to relieve acute pain. During the inpatient treatment period, the patient took herbal medicines, but no analgesics. The following four different medicines were administered at the first visit: aceclofenac (tablet, 100 mg), almagate (tablet, 500 mg), afloqualone (tablet, 20 mg), and acetaminophen and tramadol hydrochloride (semi-tablet, 181.25 mg) (Table 2).

### 2.6. Clinical Outcomes

#### 2.6.1. Numeric Rating Scale (NRS)

The NRS score for the intensity of LBP was 10 at the initial visit, which started to decline to a score of 9 on 18 February, corresponding to day 2 of the inpatient treatment period. The pain was reduced by 50% on day 7, resulting in an NRS score of 5. At discharge, the NRS score was 2, indicating an 80% improvement in LBP compared to that at baseline. At the end of outpatient treatment, the reported NRS score was 1. These results suggest that the patient showed >90% improvement in pain intensity, from an NRS score of 10 at the initial visit to an NRS score of 1 at the end date.

As for radiculopathy, the NRS score was 10 at the initial visit, and the symptoms were relieved by 50% on day 7 of the inpatient treatment period, resulting in an NRS score of 5. Continuous improvements were achieved, with an NRS score of 3 at discharge and an NRS score of 2 at the end of outpatient treatment. These results indicated that the patient exhibited >80% improvement in radiculopathy, from an NRS score of 10 at the initial visit to an NRS score of 2 at the end date (Figure 5).

Notably, the NRS scores for LBP and radiculopathy decreased by 1 point on 17, 20, 22, 27, and 28 February, which were the dates when ultrasound-guided pharmacopuncture was administered; these data are marked with asterisks in Figure 5.

#### 2.6.2. Oswestry Disability Index (ODI), EuroQol Five Dimension Index (EQ5D), and Patient Global Impression of Change (PGIC)

During the treatment period, the ODI score improved from 84.44 at admission (17 February) to 44.44 at discharge (28 February) and 28.89 at the end of outpatient treatment (7 June). The EQ5D score also improved from 0.303 at admission (17 February) to 0.829 at discharge (28 February) and slightly increased to 0.871 at the end of outpatient treatment (7 June), indicating improvement in the patient’s quality of life by approximately 0.568 points compared to the score at the initial visit (Figure 6). With respect to the PGIC, the patient conducted a subjective evaluation of the impression of his own improvement on a 7-point scale. A questionnaire was used for evaluation at discharge and at the end of outpatient treatment, with a response of 1 (very much improved).

## 3. Discussion

In this study, with rapid improvement in symptoms as the treatment goal for a patient with an HIVD complaining of acute LBP, we evaluated a new treatment method in which an ultrasound-guided Kambin’s triangle approach to high-dose pharmacopuncture administration was adopted. Additionally, we examined the effectiveness and safety of the proposed method.

IVDs are avascular structures. Only the outer annulus fibrosus has a few blood vessels (metaphyseal arteries) remaining from embryonic development. The vascular supply to the IVDs is limited, and the nutrient supply to most disc cells is indirect, passing through the adjacent vertebral endplate. Thus, IVDs have a poor inherent healing potential [10]. The pathophysiology of discogenic pain or pain from LDH involves mechanical nerve root compression by the herniated disc or chemical irritation from injury to the discs and nerve roots [19]. In Western medicine, the common treatment modalities include physical therapy, oral medications, and injections. With respect to injections, corticosteroids are frequently used as a local treatment targeting the affected nerve roots [9]. Other frequently used conservative treatments, such as epidural steroid injections, are currently administered under C-arm fluoroscopic guidance in most cases. The ultrasound-guided technique has the advantages of being non-invasive and without radiation exposure; furthermore, it is readily accessible in clinical settings and is beneficial in terms of ensuring patient safety because it allows the procedure to be performed while visually examining the major organs, blood vessels, and nerves of the corresponding acupoints in real time. The ultrasound-guided technique also enables the utilization of a minimally invasive route based on accurate knowledge of the anatomical structures to ensure a safe approach to the target structure. Thus, when applied to pharmacopuncture, this technique provides adequate guidance for the safe and effective performance of procedures.

In traditional Korean medicine, the mechanism of action of pharmacopuncture for treating musculoskeletal disorders primarily involves the interaction between the mechanical effect of the physical stimulation (e.g., irrigation, hydrodissection) of the acupoints and the chemical effect of the injected pharmacopuncture solution [13]. For the Shinbaro 2 pharmacopuncture used in this study, four herbs were added to five medicinal herbs of GCSB-5; therefore, the solution contained nine medicinal herbs as ingredients. In vitro and in vivo studies of LDH and lumbar spinal stenosis, respectively, showed that Shinbaro 2 pharmacopuncture had an anti-inflammatory effect and suppressed disc degeneration in rat models [20,21]. With regard to the anti-inflammatory effect, the serum prostaglandin E2, interleukin-1β, and tumor necrosis factor-α levels decreased in a dose-dependent manner, and such an effect of dose-dependent reduction was also observed at the protein and mRNA levels. Analyses also showed that Shinbaro 2 administration led to a dose-dependent reduction in disc-degeneration-related factors, such as matrix metalloproteinase (MMP) 9, MMP13, and ADAMTS-5, at the protein and mRNA levels [21]. In fact, Shinbaro 2 pharmacopuncture has been reported to be actively administered for patients with an HIVD in clinical Korean medical practice [22].

The pharmacopuncture solution dose used for patients with LBP has been reported to range from 1.00 to 1.71 mL. Pharmacopuncture is mainly administered at acupoints around the lower back; nevertheless, there are also cases in which it is administered at the hip joint or lower extremities [14]. With respect to high-dose pharmacopuncture using Shinbaro 2, previous studies have reported the injection of a 4 mL dose targeting the facet joint; however, all the procedures were performed without guidance, and no study on the direct administration of high-dose (6 mL or more) pharmacopuncture into the affected peridiscal space has yet been reported [16,23]. Ryu et al. used integrated Korean medical treatment that included 4 mL Shinbaro pharmacopuncture and reported improvement in a patient with acute LDH; their results showed that it took approximately 25 days to achieve a 50% reduction in the NRS score and >3 months to achieve an almost complete resolution of pain with an NRS score close to 1 [23].

Epidural corticosteroid injections can be classified into three types: transforaminal, caudal, and interlaminar. Among these, the transforaminal approach is the most accurate and effective route of administration [24,25,26]. In this case, the frequently used supraneural approach may damage the vascular structures because the artery of Adamkiewicz is located in this area; additionally, complications such as needle-induced vascular spasms and ischemic spinal cord injury have been reported [27,28,29]. For these reasons, the use of alternative approaches such as Kambin’s triangle instead of the “safe triangle” approach has been suggested to ensure patient safety and procedural stability [30]. The Kambin’s triangle approach is used in endoscopic lumbar discectomy via the posterolateral approach. The relative safety of endoscopic foraminotomy is implicitly related to the safety of the Kambin’s triangle approach in relation to nerve root and vascular injury. Although this serves as a safe route, caution should be exercised because this approach may directly cause IVD puncture. Park et al. reported that, with respect to the location of the artery of Adamkiewicz, more than 50% of cases were located in the upper half, whereas almost no cases were located in the inferior one-fifth of the foramen. Thus, the final position of the needle should be in the inferior and posterior positions within the neural foramen to prevent direct injury to the IVD and artery during the procedure. Theoretically, the treatment effect using the Kambin’s triangle approach may be inferior to that of the supraneural approach because the final position of the needle tip lies posterior to the supraneural approach; however, in real-world evaluations, these two approaches have been reported to show no difference in terms of improvement in pain or functional disability [31]. The new pharmacopuncture procedure used in this study targeted the perineural and peridiscal spaces, not the epidural space. Therefore, Kambin’s triangle is considered an effective route to inject the pharmacopuncture solution targeting the nerve roots and IVDs in an anatomically safe manner and to achieve the significant alleviation of pain and functional disability caused by IVD and nerve root lesions. Given that an ultrasound-guided intervention based on anatomical structures enables a direct approach to the affected area of the extruded disc and nerve injury, as confirmed on MRI, a more effective action of the pharmacopuncture solution can be expected compared to the unguided procedure simply targeting the facet joint, which is the conventional method. Furthermore, the high-dose administration of 6 mL was considered to have demonstrated greater effects of irrigation and hydrodissection, along with dose-dependent pharmacological actions of Shinbaro 2, advancing beyond the conventional method of pharmacopuncture. The results of this study indicated significant improvements in three outcomes—namely, the pain scale score (NRS), functional disability index (ODI), and quality of life index (EQ5D)—and in complaints regarding the patient’s symptoms; the changes in these outcomes were remarkable during the intensive inpatient treatment period. Although the outpatient treatment did not show significant changes in the outcome measures, the patient’s symptoms continued to improve, and physical function was maintained. Although making a direct comparison is not possible due to the differences in patients’ conditions and symptoms, the rate of recovery achieved in this study was faster than that reported in a previous study, which was a case report on an acute HIVD with integrated Korean medical treatment [23]. Furthermore, the ultrasound-guided procedure through the Kambin’s triangle approach did not cause severe adverse events from contact with the nerve root or injury to the tissue.

This case report illustrates the promising benefits of ultrasound-guided pharmacopuncture for managing an acute HIVD with high-dose injections. The key lessons learned include the enhanced precision and safety afforded by ultrasound guidance, which facilitates the accurate targeting of the perineural and peridiscal spaces and minimizes complications. The high-dose Shinbaro 2 pharmacopuncture demonstrated significant pain relief, functional improvement, and improved quality of life, suggesting its superior therapeutic potential. The rapid symptom relief observed during the intensive inpatient treatment period underscores the efficacy of this integrative approach. This case supports the further exploration and integration of traditional Korean medicine with modern diagnostic and therapeutic techniques.

## 4. Conclusions

This case report demonstrated the potential of introducing safe and effective ultrasound-guided high-dose spinal pharmacopuncture, targeting the perineural space rather than the epidural space. The strengths of this study include the presentation of an ultrasound-guided intervention that offers precise and direct targeting, enhancing the safety and effectiveness of the treatment. The use of high-dose Shinbaro 2 pharmacopuncture showed significant therapeutic benefits, including rapid symptom relief and functional improvement.

The patient reported a notable decrease in pain levels, with the numeric rating scale (NRS) for pain reducing from 10 to 2 by the end of the inpatient treatment, and further down to 1 by the end of outpatient follow-up. Similarly, the Oswestry disability index (ODI) improved from 84.44 at admission to 28.89 at the end of treatment, and the EuroQol-5D (EQ-5D) score increased from 0.303 to 0.871, indicating a significant enhancement in quality of life. These results suggest that high-dose Shinbaro 2 pharmacopuncture can effectively manage symptoms and improve patient outcomes.

However, this study has limitations, such as the small sample size, being a single-case report, and the lack of direct comparison with existing methods of general pharmacopuncture or conventional (Western) medicine injection therapies. Future research should focus on designing and conducting multicenter randomized controlled trials with larger sample sizes to provide high-quality evidence for the safety and effectiveness of this new pharmacopuncture procedure.

## Figures and Tables

**Figure 1 healthcare-12-01620-f001:**
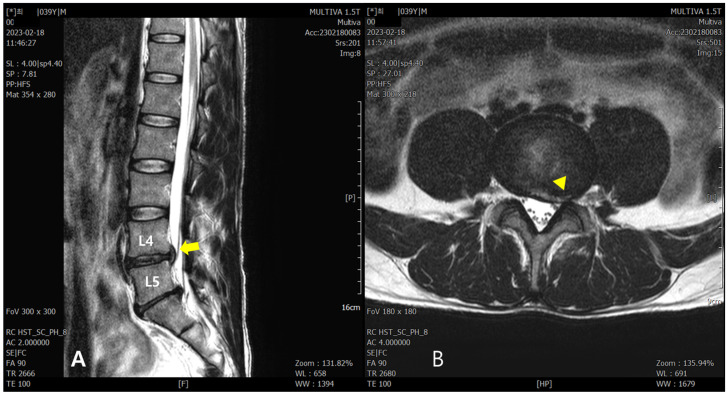
Lumbar spine MRI conducted on 18 February 2023. (**A**) T2-weighted turbo-spin-echo sagittal view: the arrow indicates the area of the extruded L4/5 disc, and the L5 nerve roots are abutted against the extruded disc. (**B**) T2-weighted turbo-spin-echo axial view: the triangle indicates the area where the disc has extruded from the central to left foraminal zone.

**Figure 2 healthcare-12-01620-f002:**
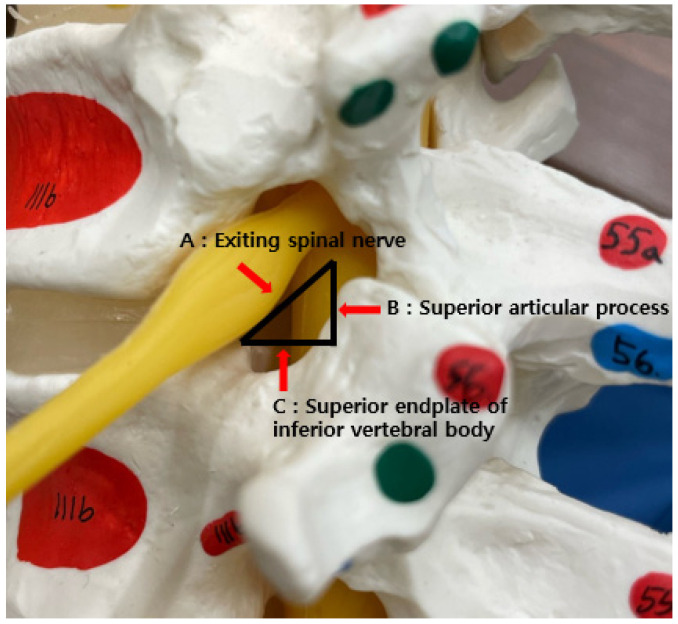
Kambin’s triangle (yellow line) bordered by the exiting nerve root ((**A**) hypotenuse), thecal sac ((**B**) height), and superior border of the caudal vertebra ((**C**) width).

**Figure 3 healthcare-12-01620-f003:**
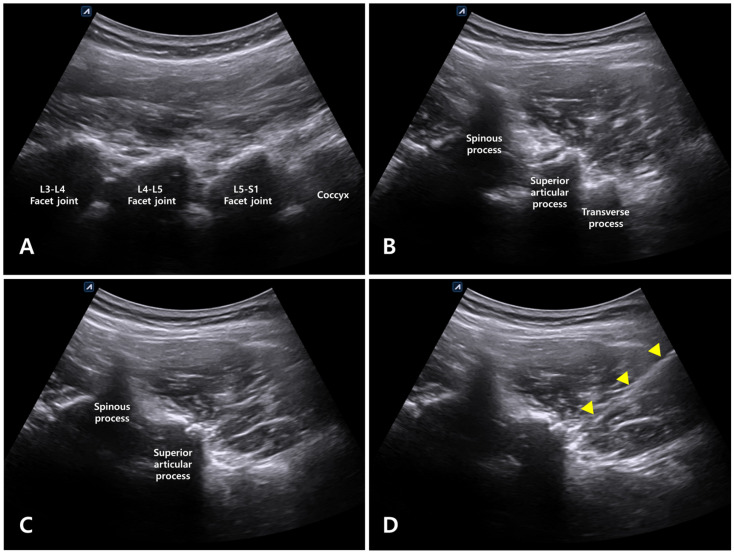
Ultrasound-guided Kambin’s triangle approach. (**A**) In the long-axis view, the facet joint was confirmed via the camel-hump sign. (**B**) In the short-axis view, the superior articular process of the L4–L5 facet joint, L5 transverse process, and L4 spinous process were observed in one plane. (**C**) The probe was moved cephalad, the transverse process disappeared, and only the superior articular process and facet joint were observed. (**D**) In the same plane as Figure (**C**), the path through which the needle entered Kambin’s triangle (marked with triangles) is shown.

**Figure 4 healthcare-12-01620-f004:**
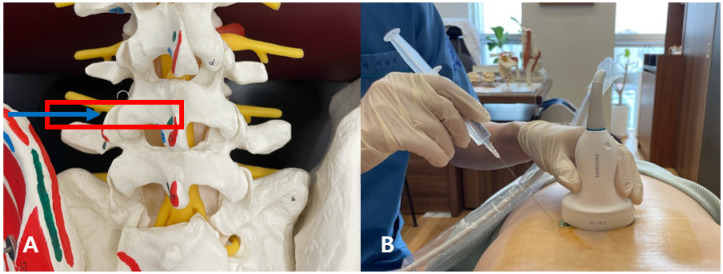
Actual scan area and needle direction during the ultrasound-guided intervention. (**A**) The location of the ultrasound probe in the spine model. The square denotes the probe location and the arrow indicates the needle entry direction. (**B**) The position of the probe and the entry angle of the injection needle during the actual procedure.

**Figure 5 healthcare-12-01620-f005:**
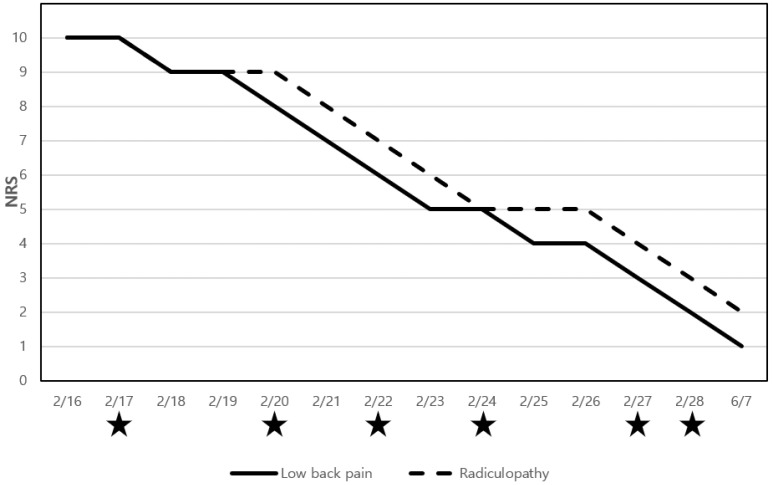
Numeric rating scale scores for low back pain and radiculopathy. The asterisk indicates the date on which the ultrasound-guided procedure was performed.

**Figure 6 healthcare-12-01620-f006:**
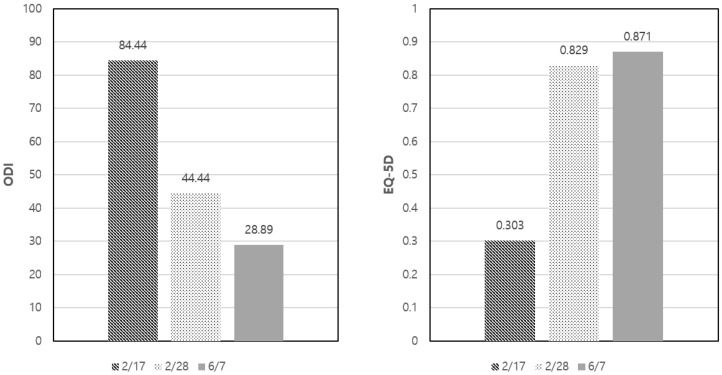
Changes in the ODI and EQ5D scores on the admission date (2/17), discharge date (2/28), and end date of outpatient treatment (6/7). ODI, Oswestry disability index; EQ5D, EuroQol five-dimension index.

**Table 1 healthcare-12-01620-t001:** Timeline of ultrasound-guided pharmacopuncture and Korean medicine treatment with changes in pain and symptom evaluation indicators of acute disc herniation pain.

	Outpatient	Inpatient												Outpatient
Date	16/02	17/02	18/02	19/02	20/02	21/02	22/02	23/02	24/02	25/02	26/02	27/02	28/02	07/06
Ultrasound-guided PA-Tx	-	O	-	-	O	-	O	-	O	-	-	O	O	-
PA-Tx	O	O	O	O	O	O	O	O	O	O	O	O	O	O
A-Tx	O	O	O	O	O	O	O	O	O	O	O	O	O	O
EA	O	O	O	O	O	O	O	O	O	O	O	O	O	O
Cupping	O	O	O	O	O	O	O	O	O	O	O	O	O	O
Chuna	-	O	O	O	O	O	O	O	O	-	-	O	O	O
H-med	O	O	O	O	O	O	O	O	O	O	O	O	O	O
P-Tx	-	O	O	-	O	O	O	-	-	O	-	O	O	-
W-med	O	-	-	-	-	-	-	-	-	-	-	-	-	-

Abbreviations: PA-Tx, pharmacopuncture treatment; A-Tx, acupuncture treatment; EA, electroacupuncture; H-med, herbal medicine; P-Tx, physical therapy; W-med, Western medicine.

**Table 2 healthcare-12-01620-t002:** Types of pharmacopuncture used in this study.

	Ultrasound-Guided	Non-Ultrasound-Guided
Pharmacopuncture procedure name	Shinbaro 2	
Ingredients	Rhizomes of *Cibotium barometz* (0.0013 g/mL), roots of *Saposhnikovia divaricata* (0.0013), stem barks of *Eucommia ulmoides* (0.0013), stems and roots of *Acanthopanax sessiliflorus* (0.0013), rhizomes and roots of *Ostericum koreanum* (0.0013), roots of *Angelica pubescens* (0.0013), roots of *Achyranthes japonica* (0.0013), *Scolopendra subspinipes* (0.0013), roots of *Paeonia albiflora* (0.0027)	
Volume	6 mL per point	1 mL per point
Needle size	26 gauge × 90 mm	29 gauge × 13 mm
Location of needle insertion	Kambin’s triangle	Quadratus lumborum and erector spinae muscles (Ashi points)

## Data Availability

The data presented in this study are available from the corresponding author upon request.

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
