# Peer review of "Integrative Medicine Focusing on Ultrasound-Guided High-Dose Shinbaro 2 Pharmacopuncture for Acute Herniated Intervertebral Discs: A Case Report"

_healthcare, 2024, doi:10.3390/healthcare12161620_

Round 1

Reviewer 1 Report

Comments and Suggestions for Authors

This manuscript reports a case about ultrasound-guided high dose shinbaro 2 pharmacopuncture for acute herniated intervertebral discs. The paper describes in detail the patient’s condition and the course of treatment. Before publishing, the following issues should be noted.

1. In the introduction part, line 82, the authors claimed “No previous study has examined the treatment effects of ultrasound-guided high dose Shinbaro 2 pharmacopuncture in patients with LDH”. Please state this carefully. Human knowledge is limited.

Wang, Ye-hui MDa,*; Zhou, Yi MDa; Xie, Yi-zhou MDa; Fan, Xiao-hong PhDa; Liang, Wan-qiang MDb; Wei, Xing MDb; Zhao, Ming-dong MDb; Huo, Yu-xiong BDb; Zhang, Ting BDb; Yin, Yun PhDb The effect of ultrasound-guided acupotomy and Juanbi decoction on lumbar disc herniation: A randomized controlled trial. Medicine 102(1):p e32622, January 06, 2023. | DOI: 10.1097/MD.0000000000032622

2. In Figure 1, icon B disappeared in the second picture.

3. In Figures 5 and 6, vertical scales should be added.

4. As evidence of the effectiveness of the treatment, whether the patient undergo radiology and ultrasound examination after Feb 28, 2023 ?

5. The discussion part is logically confused, especially the last paragraph (line 394-400), and should belong in the conclusion part. The expression of this paragraph is chaotic.

6. As an academic paper, the conclusion part should include the important results of the paper. Obviously, the current conclusion section is too short and does not contain the main conclusion of the paper. The conclusions part should be rewritten.

Best wishes

Author Response

Dear reviewer 1,

We appreciate your detailed comments and also for understanding our opinions about some of the previous comments. Although there are still some shortcomings, we think that your suggestions were extremely helpful in improving our paper. In the following sections, please find our responses to each of your comments and suggestions. All changes were colored in green in the revised manuscript. We hope our answers and revisions have clarified many of your questions and concerns.

  1. In the introduction part, line 82, the authors claimed “No previous study has examined the treatment effects of ultrasound-guided high dose Shinbaro 2 pharmacopuncture in patients with LDH”. Please state this carefully. Human knowledge is limited.

Wang, Ye-hui MDa,*; Zhou, Yi MDa; Xie, Yi-zhou MDa; Fan, Xiao-hong PhDa; Liang, Wan-qiang MDb; Wei, Xing MDb; Zhao, Ming-dong MDb; Huo, Yu-xiong BDb; Zhang, Ting BDb; Yin, Yun PhDb The effect of ultrasound-guided acupotomy and Juanbi decoction on lumbar disc herniation: A randomized controlled trial. Medicine 102(1):p e32622, January 06, 2023. | DOI: 10.1097/MD.0000000000032622

  • Response: Thank you for the valuable suggestion. We have revised the statement in the manuscript to acknowledge the limitations of current knowledge. The revised sentence now reads: “Previous research has explored ultrasound-guided acupotomy and herbal medicine for LDH, but no previous study has specifically examined the treatment effects of ultrasound-guided high-dose Shinbaro 2 pharmacopuncture in patients with LDH, to the best of our knowledge.”

Regarding the paper by Wang et al. (2023), it is important to note that their study focuses on the effect of ultrasound-guided acupotomy and Juanbi decoction on lumbar disc herniation, which differs from our study that examines the effects of pharmacopuncture. While both studies employ ultrasound guidance, the treatment modalities (acupotomy vs. pharmacopuncture) and the substances used (Juanbi decoction vs. Shinbaro 2) are different. Specifically, Juanbi decoction is an oral medication, whereas Shinbaro 2 is an injectable solution. We appreciate the reference and have considered it in the context of our research.

Thank you for your constructive feedback, which has been invaluable in enhancing our paper.

  1. In Figure 1, icon B disappeared in the second picture.
  • Response: Thank you for your meticulous observation regarding Figure 1. We have addressed this issue by ensuring that icon B is now clearly visible in the second picture. We appreciate your attention to detail, which has helped us enhance the clarity and consistency of our visual presentation.

  1. In Figures 5 and 6, vertical scales should be added.
  • Response: Thank you for your valuable suggestion concerning Figures 5 and 6. We have added vertical scales to both figures to improve their clarity and interpretability. This modification ensures that the data and trends are accurately conveyed, thereby enhancing the overall quality of our manuscript.

  1. As evidence of the effectiveness of the treatment, whether the patient undergo radiology and ultrasound examination after Feb 28, 2023 ?
  • Response: Thank you for your insightful comment. Unfortunately, follow-up radiology and ultrasound examinations were not performed after February 28, 2023, for this patient. However, we have provided comprehensive clinical outcomes based on the patient's subjective reports and objective measures such as the Numeric Rating Scale (NRS), Oswestry Disability Index (ODI), and EuroQol-5D (EQ-5D) scores, which showed significant improvement. These clinical evaluations provide substantial evidence of the treatment's effectiveness.

Additionally, this study primarily focuses on evaluating the clinical symptoms of the patient through ultrasound-guided interventions rather than comparing the pre- and post-treatment status of the disc itself. Ultrasound was utilized solely as a guiding tool for the pharmacopuncture procedure and is limited in its capacity to evaluate the disc condition for comparative purposes. Therefore, it is not suitable for use as a pre- and post-comparative tool in this context.

Nonetheless, we acknowledge that not performing a follow-up MRI is a limitation and could be seen as a missed opportunity to provide additional objective evidence of the treatment's effectiveness. We recognize the importance of follow-up imaging studies and will consider incorporating them in future research to further validate the treatment outcomes.

We appreciate your constructive feedback and believe that addressing this aspect in future research will significantly improve the quality and robustness of our studies. Thank you once again for your valuable insights and guidance.

  1. The discussion part is logically confused, especially the last paragraph (line 394-400), and should belong in the conclusion part. The expression of this paragraph is chaotic.
  • Response: Thank you for your insightful comments. Please note that a detailed response to this comment is included with our response to Comment 6.
  1. As an academic paper, the conclusion part should include the important results of the paper. Obviously, the current conclusion section is too short and does not contain the main conclusion of the paper. The conclusions part should be rewritten.
  • Response: Thank you for your insightful comments. We have carefully considered your feedback and made the necessary revisions to improve the clarity and logical flow of our manuscript.

Regarding Comment 5, we have addressed the logical confusion in the discussion section by integrating the last paragraph (lines 394-400) into the conclusion. This adjustment provides a coherent summary of the study’s strengths, limitations, and future directions.

Regarding Comment 6, we have expanded the conclusion to include the important results of the paper, ensuring it comprehensively reflects the main findings and conclusions. The revised conclusion now reads:

======================================================================

Conclusion

This case report demonstrated the potential of introducing safe and effective ultrasound-guided high-dose spinal pharmacopuncture, targeting the perineural space rather than the epidural space. The strengths of this study include presenting an ultrasound-guided intervention that offers precise and direct targeting, thereby enhancing the safety and effectiveness of the treatment. The use of high-dose Shinbaro 2 pharmacopuncture showed significant therapeutic benefits, including rapid symptom relief and functional improvement.

The patient reported a notable decrease in pain levels, with the Numeric Rating Scale (NRS) for pain reducing from 10 to 2 by the end of the inpatient treatment, and further down to 1 by the end of outpatient follow-up. Similarly, the Oswestry Disability Index (ODI) improved from 84.44 at admission to 28.89 at the end of treatment, and the EuroQol-5D (EQ-5D) score increased from 0.303 to 0.871, indicating a significant enhancement in quality of life. These results suggest that high-dose Shinbaro 2 pharmacopuncture can effectively manage symptoms and improve patient outcomes.

However, this study has limitations, including the small sample size, being a single-case report, and the lack of direct comparison with existing methods of general pharmacopuncture or conventional (Western) medicine injection therapies. Future research should focus on designing and conducting multicenter randomized controlled trials with larger sample sizes to provide high-quality evidence for the safety and effectiveness of this new pharmacopuncture procedure. These further studies will help validate our findings and establish robust evidence for integrating traditional Korean medicine with modern diagnostic and therapeutic techniques.

======================================================================

We hope these revisions address your concerns and improve the manuscript's overall quality and clarity. Thank you again for your constructive feedback, which has been invaluable in enhancing our paper.

Reviewer 2 Report

Comments and Suggestions for Authors

The case report is very interesting but some data are missed.

1. Please provide the ethical approval number from IRB in the introduction not only at the end.

2. In figure 1, the second MRI (axial view) miss letter (B) on the picture itself.

3. Did you evaluated the pain of the patient with the BPI numerical scale to measure the pain intensity during the physical examination?

4. The dates are confusing.  On February 17, he visited the Rehabilitation Department of 99 Traditional Korean Medicine at our hospital. then, you stated that The patient visited the hospital for an initial consultation on February 16 and the MRI was on 18. That means the treatment was prescribed before the MRI?! clarify please.

5. the Pharmacopuncture was described in a good details. I just suggest to address any symptoms felt by the patient after this technique.

6. please clarify if the Pharmacopuncture needs anaesthesia?

7. You have to clarify that the recorded scales are descriptive statistics for the changes in the ODI and EQ5D scores on the admission date (2/17), discharge date (2/28), 294 and end date of outpatient treatment (6/7).

8. Discussion and conclusion were clear and to the point. 

9. would you provide an empty form of the obtained informed consent from the patient, not for publication but only to check.

Author Response

Dear reviewer 2,

We appreciate your detailed comments and also for understanding our opinions about some of the previous comments. Although there are still some shortcomings, we think that your suggestions were extremely helpful in improving our paper. In the following sections, please find our responses to each of your comments and suggestions. All changes were colored in green in the revised manuscript. We hope our answers and revisions have clarified many of your questions and concerns.

  1. Please provide the ethical approval number from IRB in the introduction not only at the end.
  • Response: Thank you for the valuable suggestion. We have added the ethical approval number from the IRB in the introduction section to ensure clarity and compliance with ethical standards throughout the manuscript.

  1. In figure 1, the second MRI (axial view) miss letter (B) on the picture itself.
  • Response: Thank you for pointing out this detail. We have corrected Figure 1 to include letter (B) on the second MRI (axial view), ensuring accuracy and completeness in our visual representation.

  1. Did you evaluated the pain of the patient with the BPI numerical scale to measure the pain intensity during the physical examination?
  • Response: Thank you for the good point. Unfortunately, we only used the NRS and did not use the BPI numerical scale. The NRS was used to evaluate pain intensity in order to quickly and easily assess pain without causing additional burden to the patient, especially in situations where the pain is severe. This method allows for rapid evaluation, ensuring that the patient's discomfort is minimized during the assessment process.

In contrast, the BPI (Brief Pain Inventory) can evaluate pain across multiple sites, track changes in pain levels, and assess the broader impact of pain on a patient's life, providing a more comprehensive assessment. Future studies should incorporate tools like the BPI to enhance the accuracy of pain evaluation.

  1. The dates are confusing. On February 17, he visited the Rehabilitation Department of 99 Traditional Korean Medicine at our hospital. then, you stated that The patient visited the hospital for an initial consultation on February 16 and the MRI was on 18. That means the treatment was prescribed before the MRI?! clarify please.
  • Response: Thank you for the good point. We apologize for the confusion. We have clarified the dates in the manuscript to ensure they are consistent and logical. On February 16, during the initial consultation, general treatment was administered. On the admission day, February 17, one session of ultrasound-guided pharmacopuncture was performed before the MRI on February 18 to control the pain. This was based on the L5 dermatome symptoms, and the potential lesion area was evaluated and treated in the L4/L5 Kambin's triangle region.

In clinical practice, it is often necessary to address the patient's symptoms immediately based on clinical evaluation before imaging studies can be performed. Therefore, one session of ultrasound-guided pharmacopuncture was performed based on clinical symptoms prior to the MRI. We have specified this in the manuscript in section 2.5.2 of the manuscript. The revised text now includes: "The first session, conducted on February 17, utilized the Kambin's triangle approach, based on clinical symptoms, to target the L4/L5 region prior to the MRI on February 18. Subsequent sessions, following MRI confirmation of the L4/L5 disc lesion, continued to use the Kambin's triangle approach for precise treatment of the affected area."

We appreciate your attention to detail, which has helped us improve the clarity and accuracy of our manuscript. Thank you again for your valuable insights and guidance.

  1. the Pharmacopuncture was described in a good details. I just suggest to address any symptoms felt by the patient after this technique.
  • Response: Thank you for your insightful suggestion. We have addressed your comment by adding a detailed description of the symptoms experienced by the patient during the ultrasound-guided pharmacopuncture procedure to section 2.5.2 of the manuscript. The revised text now includes: “During the procedure, the patient experienced localized tightness and heaviness at the initial drug injection site, and as the amount of drug increased, a tightening and heavy sensation gradually developed along the L5 dermatome.”

We appreciate your valuable feedback, which has helped us improve the comprehensiveness and clarity of our manuscript. Thank you again for your constructive comments.

  1. please clarify if the Pharmacopuncture needs anaesthesia?
  • Response: Thank you for your valuable question. We have clarified the use of anesthesia in the pharmacopuncture procedure by adding the following details to section 2.5.2 of the manuscript: “A 26-gauge needle, which is relatively fine and causes minimal tissue damage, was used for the procedure. Additionally, the slight tingling sensation typically felt during pharmacopuncture is generally within the tolerable range for patients. Therefore, in clinical practice, anesthesia was not often used during the treatment.”

We appreciate your insightful suggestion, which has helped us improve the clarity and comprehensiveness of our manuscript. Thank you again for your constructive comments.

  1. You have to clarify that the recorded scales are descriptive statistics for the changes in the ODI and EQ5D scores on the admission date (2/17), discharge date (2/28), 294 and end date of outpatient treatment (6/7).
  • Response: Thank you for your valuable suggestion. We have clarified the use of recorded scales and included the specific dates as you recommended. The revised text in section 2.6.2 now reads: “During the treatment period, the ODI score improved from 84.44 at admission (February 17) to 44.44 at discharge (February 28) and 28.89 at the end of outpatient treatment (June 7). The EQ5D score also improved from 0.303 at admission (February 17) to 0.829 at discharge (February 28) and slightly increased to 0.871 at the end of outpatient treatment (June 7), indicating improvement in the patient’s quality of life by approximately 0.568 points compared to the score at the initial visit (Figure 6). With respect to the PGIC, the patient conducted a subjective evaluation of the impression of his own improvement on a 7-point scale. A questionnaire was used for evaluation at discharge and at the end of outpatient treatment, with a response of 1 (very much improved).”

We appreciate your insightful suggestion, which has helped us improve the clarity and comprehensiveness of our manuscript. Thank you again for your constructive comments.

  1. Discussion and conclusion were clear and to the point.
  • Response: Thank you for your kind words regarding our discussion and conclusion sections. We are deeply grateful for your valuable insights throughout the review process. This experience has been incredibly enlightening, and we have learned and grown significantly while preparing this manuscript. Your constructive feedback has greatly enhanced the quality of our work. Thank you once again for your outstanding guidance and support.

  1. would you provide an empty form of the obtained informed consent from the patient, not for publication but only to check.
  • Response: Thank you for the good point. We will provide an empty form of the obtained informed consent from the patient for your review (please see the attached file).

Reviewer 3 Report

Comments and Suggestions for Authors

The authors have developed a well-conducted and well-written case report
aimed to explore the effectiveness of an integrative medicine approach,
specifically focusing on ultrasound-guided high-dose Shinbaro 2
pharmacopuncture, for treating acute herniated intervertebral discs (HIVD)
that cause low back pain and radiating pain.

However, I would like to make a few observations before recommending their
work for publication.

1. The CARE (Case Report) criteria are designed to improve the quality of
case reports and ensure comprehensive and transparent reporting of patient
cases in the medical literature. These criteria encompass various aspects of
reporting, including the patient's demographic information, diagnosis,
interventions, outcomes, and lessons learned from the case.  Your paper does
not explicitly mention the use of the CARE criteria.

2. I encourage the authors to comment further, although briefly, on
conservative treatment. 

3. Why didn't the authors think to evaluate their patients' expectations or
verbal suggestions? I advise the authors to discuss this, perhaps as part of
a future study with larger sample size. I leave you with two useful works
from which to draw inspiration:
    doi.org/10.47197/retos.v46.93950
    DOI: 10.3390/ijerph18084206

4. Could you add a section on "Lessons Learned"?”

Comments on the Quality of English Language

No comments

Author Response

Dear reviewer 3,

We appreciate your detailed comments and also for understanding our opinions about some of the previous comments. Although there are still some shortcomings, we think that your suggestions were extremely helpful in improving our paper. In the following sections, please find our responses to each of your comments and suggestions. All changes were colored in green in the revised manuscript. We hope our answers and revisions have clarified many of your questions and concerns.

  1. The CARE (Case Report) criteria are designed to improve the quality of case reports and ensure comprehensive and transparent reporting of patient cases in the medical literature. These criteria encompass various aspects of reporting, including the patient's demographic information, diagnosis, interventions, outcomes, and lessons learned from the case. Your paper does not explicitly mention the use of the CARE criteria.
  • Response: Thank you for the valuable suggestion. We have addressed your comment by explicitly mentioning the use of the CARE criteria in the introduction. Additionally, we have thoroughly revised the manuscript to ensure comprehensive and transparent reporting of almost all aspects of the CARE criteria. This includes the patient's demographic information (detailed in section 2.1. Clinical Features), diagnosis (detailed in section 2.4. Diagnosis), interventions (detailed in section 2.5. Treatment), outcomes (detailed in section 2.6. Clinical Outcomes), and lessons learned (detailed in section 5. Lessons Learned).

We will also be submitting the completed CARE checklist along with the revised manuscript for your review.

We appreciate your guidance, which has significantly enhanced the quality and clarity of our case report. Thank you once again for your insightful comments.

  1. I encourage the authors to comment further, although briefly, on conservative treatment.
  • Response: Thank you for your valuable suggestion. We have addressed your comment by adding a brief explanation of conservative treatment in the introduction section of the manuscript. The revised text now includes: “Conservative treatment refers to non-surgical methods used to manage and treat medical conditions, including acute HIVD. These treatments are aimed at reducing symptoms, improving function, and promoting recovery without the need for invasive procedures.”

We appreciate your insightful suggestion, which has helped us improve the clarity and comprehensiveness of our manuscript. Thank you again for your constructive comments.

  1. Why didn't the authors think to evaluate their patients' expectations or verbal suggestions? I advise the authors to discuss this, perhaps as part of a future study with larger sample size. I leave you with two useful works from which to draw inspiration: doi.org/10.47197/retos.v46.93950 DOI: 10.3390/ijerph18084206
  • Response: Thank you for the valuable suggestion. We did not evaluate the patients' expectations or verbal suggestions in this study. However, we acknowledge the importance of such evaluations and will consider incorporating them in future studies with larger sample sizes. We appreciate the references provided and will review them for insights on how to better integrate patient expectations into our research.

  1. Could you add a section on "Lessons Learned"?”
  • Response: Thank you for your insightful suggestion. We have indeed learned and gained a great deal from this case study. Based on your feedback, we have added a new paragraph addressing the lessons learned to the discussion section of the manuscript. Although we did not create a separate title for this section, as the journal Healthcare specifies the structure to include Introduction, Materials and Methods, Results, Discussion, and Conclusion, we have incorporated the "Lessons Learned" content at the end of the discussion section. The paragraph reads as follows:

======================================================================

Lessons Learned

This case report illustrates the promising benefits of ultrasound-guided pharmacopuncture for managing acute HIVD with high-dose injections. Key lessons learned include the enhanced precision and safety afforded by ultrasound guidance, which facilitates accurate targeting of the perineural and peridiscal spaces and minimizes complications. The high-dose Shinbaro 2 pharmacopuncture demonstrated significant pain relief, functional improvement, and improved quality of life, suggesting its superior therapeutic potential. The rapid symptom relief observed during the intensive inpatient treatment period underscores the efficacy of this integrative approach. This case supports further exploration and integration of traditional Korean medicine with modern diagnostic and therapeutic techniques.

======================================================================

We are grateful for your constructive feedback, which has significantly enriched our manuscript. Thank you again for your valuable insights and guidance.